# Exploring Doctors’ Emerging Commitment to Rural and General Practice Roles over Their Early Career

**DOI:** 10.3390/ijerph182211835

**Published:** 2021-11-11

**Authors:** Matthew McGrail, Belinda O’Sullivan, Tiana Gurney, Diann Eley, Srinivas Kondalsamy-Chennakesavan

**Affiliations:** 1Rural Clinical School, The University of Queensland, Rockhampton 4700, Australia; 2Rural Clinical School, The University of Queensland, Toowoomba 4350, Australia; belinda.osullivan@uq.edu.au (B.O.); t.gurney@uq.edu.au (T.G.); s.kondalsamychennakes@uq.edu.au (S.K.-C.); 3Office of Medical Education, The University of Queensland, Herston 4006, Australia; d.eley@uq.edu.au

**Keywords:** rural health, general practice, career choices, early career doctors, health policy, workforce shortages

## Abstract

Producing enough doctors working in general practice or rural locations, or both, remains a key global policy focus. However, there is a lack of evidence about doctors’ emerging commitment to these decisions. This study aimed to explore changes in the level of certainty about career interest in working in general practice and working rurally, as doctors pass through various early career stages. The participants were 775 eligible respondents to a 2019 survey of medical graduates of The University of Queensland from 2002–2018. Certainty levels of specialty choice were similar between GPs and specialists up until the beginning of registrar training. At that point, 65% of GPs compared with 80% of other specialists had strong certainty of their specialty field. Consistently (and significantly) less of those working rurally had strong certainty of the location where they wanted to practice medicine at each career time point. At the start of registrar training, a similar gap remained (strong certainty: 51% rural versus 63% metropolitan). This study provides new evidence that career intent certainty is more delayed for the cohort choosing general practice and rural practice than the other options. The low level of certainty in early career highlights the importance of regular positive experiences that help to promote the uptake of general practice and rural practice.

## 1. Introduction

Increasing the proportion of doctors who specialise in general practice (primary care), work rurally, or both, remain key global policy goals [1,2]. Firstly, primary care doctors play an important contribution to addressing most of the population’s health needs, as an effective and efficient way to improve health outcomes over relying on specialised services [3]; however, in many countries a high proportion of recent graduates are opting to pursue other specialties [4,5,6]. Secondly, equity of access to medical care for all populations, regardless of where people live, is a common goal [7,8]; however, most rural populations globally have less doctors than they need [9]. The main way to effect changes in these outcomes is to understand the dynamics of decisions around specialty and the location of work in a doctor’s early career, as the likelihood of changes to these both decrease with age [10].

Many of the potential influences of specialty and practice location, targeting an increased uptake of general practice and rural location, are well researched. They are often the result of early exposures to general practice or rural practice, having positive role models or some pre-medicine intent/interest in each, attraction to potential lifestyle advantages, and an orientation to continuity of care [11,12,13,14]. However, the evidence is less clear around the timing of these decisions across a doctor’s early career training and work cycle, particularly in terms of how decision certainty progresses over time from being an intent or preference (but not yet with great commitment or certainty), to a finalised decision (with strong commitment or certainty). Moreover, it is unclear what influences or accelerates these decisions at different career points (e.g., demographics, consultants, role models, clinical experiences). Stronger understanding of the emergent patterns of interest and how the level of certainty is strengthened, as well as what influences career decisions, could enable more timely and targeted support to facilitate the growth of doctors committed to working in general practice or in rural locations.

It is likely that the trajectory of certainty about specialty and location somewhat mirror each other. It is recognised that ‘generalist’ specialties (typically, general practice or family medicine) are more conducive to working rurally [15]. For example, approximately 55% of Australia’s rural doctors are GPs, compared with 37% of metropolitan doctors. Similarly, 26% of Australia’s GPs work in rural locations compared with only 15% of other (non-GP) specialties working rurally [16]. Thus, it is to be expected that these key career decisions are related and the delays of one decision may directly delay the other. Similarly, it is to be expected that some early career doctors will have a strong long-term interest in living in rural locations (or metropolitan locations), which is likely to help shape their specialty preference. Evidence, to date, has not examined the trajectory of both of these career decisions in tandem.

Recent qualitative evidence from studies about early career doctors reveals that early career doctors go through a complex and dynamic process in determining their chosen specialty [17]; however, quantitative evidence about this process could be complementary for informing policy. Most evidence about specialty decisions and their timing is largely centred on medical students, demonstrating how specialty choice intention can vary year-to-year over the course of medical school training [18,19,20,21,22]. An Australian study found that 76% of medical students changed their first preference specialty from commencement to exit from medical school [23]. The focus on medical school relates to systems like the USA, with immediate entry of graduates into residency and specialty programs. However, evidence also suggests that there can be a high attrition rate from these programs over the 3–6 years of training as resident trainees realise their own mismatch to the specialty [24]. In settings like the UK and Australia, specialty decisions (and specialty college entry) are finalised after medical school, whilst doctors concurrently navigate employment and specialty training options each year. These junior doctors are often encouraged to experience many different specialty roles and departments during these early postgraduate years, which perhaps reassures delaying the need to lock in specialty decisions. Junior doctors in the UK note that “you don’t want to get that [decision] wrong” [25].

Evidence relating to practice location decision-making has identified factors and strategies associated with more junior doctors who are practising rurally [26]. Primarily, an increased selection of rural origin students and increased training time (immersion of at least one clinical year, both residing continuously and gaining clinical experiences) in rural areas during medical school are key strategies linked to producing more rural doctors [27,28,29]. The impact of this increases incrementally with increasing rural training time and can be enhanced if doctors experience both general practice and hospital experiences [30]. Equally, training in rural longitudinal integrated clerkships in small rural towns provides opportunities for emerging doctors to experience rural immersion, with continuity to place where they can follow patients and clinicians between hospital and the community and learn about longitudinal care under the supervision of rural generalists [31,32]. Some evidence demonstrates the association between early rural exposures and an ‘outcome’ of rural practice intent whilst still at medical school [33,34]. Similarly, location intent at medical school exit is much more strongly translated to rural practice compared with intent at entry [35]. It is implicitly assumed that rural origin or rural training will independently drive junior doctors towards rural practice, but this ignores the complexity of work, training and life exposures that occur in a doctor’s early career, particularly once doctors leave the structured environment of medical schools. There is a distinct lack of evidence about the emergence of certainty about where doctors want to work over the course of their early career beyond medical school.

The aim of this study is to explore changes in the level of certainty about career interest in working in general practice and working rurally, as doctors pass through various early career stages. Secondly, this study explores what exposures influenced these career interests. The study is undertaken in the Australian system, where junior doctors can have good control of the trajectory of their career but are often limited by employment and training markets.

## 2. Materials and Methods

Listings of all University of Queensland (UQ) medical school graduates between 2002 and 2018 were obtained from UQ’s administrative dataset and were then matched with Australian Health Practitioner Regulation Agency (AHPRA) registrations as of 2019 (thus ranging between their 1st to 17th postgraduate year). After excluding international students, all matched graduates (n = 4540) were invited via their last known email address (78% of these were their UQ student email addresses) to complete a 44-item online survey about postgraduate work and training in 2019. Questions were developed, including pilot testing with 5–6 doctors, to meet the project’s objective to better understand the career decisions and outcomes of UQ medical graduates. Non-respondents were given two reminders. This study had approval (numbers 2018001630 and 2012001171) from The University of Queensland Human Research Ethics Committee.

### 2.1. Context of this Study

Firstly, the UQ medical program is four-year-long postgraduate medical training program, predominantly delivered in Brisbane (major city), although of the same curriculum irrespective of training location. Our study period contains a mix of MBBS (Bachelor of Medicine Bachelor of Surgery) and MD (Doctor of Medicine) graduates. UQ’s Rural Clinical School (RCS) pathway has four teaching sites, each based in the large regional hospitals of communities of 50,000–130,000 population and located around 135–630 km from Brisbane. Training at the RCS sites occurs for a whole academic year, in one, or both, of years 3 and 4 of the course (up to two years). Additionally, over 90% of UQ domestic medical students, irrespective of RCS training, complete 6–8 weeks experience commonly in general practice in year 3 in small rural towns under UQ’s Rural and Remote Medical program. 

Secondly, the pathways for doctors who graduate from medicine in Australia are often complex with a high degree of uncertainty [36,37]. Medical graduates begin with two pre-registrar years working mostly in larger hospitals, including completing their internship. There are few opportunities for pre-registrar training experience in smaller rural hospitals or in the community (e.g., general practice). Depending on their specialty preference, they may then begin to apply (competitively) for specialty college positions, though longer pre-registrar periods are very common. Specialist training (working as a registrar) spans around 3–6 years. Through this whole postgraduate training pathway, most employment contracts are yearly and changing work location is common.

### 2.2. Comparison Groups—Specialty and Practice Location

Specialty was self-reported by registrars (enrolled in specialty training) and consultants (who had completed all specialty training and qualified as fellows) and aggregated as being either GP (‘General practice’ or ‘Rural and remote area medicine’) or non-GP (all other specialties). Doctors not yet enrolled in specialty training (pre-registrar) were excluded from analyses relating to specialty choice.

Practice location was self-reported as town/suburb and postcode, defined by their current work location. Location was then defined under the Modified Monash Model (MMM) national classification as rural for MMM 2–7 communities or metropolitan for MMM-1 [38].

### 2.3. Key Outcomes

Participants were asked to reflect back on five career stages (start of medical school, end of medical school, after internship, start of registrar training, end of registrar training), each of which align with distinctive points of Australia’s training pathway, and for each stage, answer the following questions: How certain were you about where you wanted to practice medicine (geographic location)? and How certain were you about which field of medicine (specialty) you wanted to practice medicine? Participants reported their ‘certainty’ on a 10-point scale from 1 (no idea) through to 10 (absolutely certain). Scores of 8–10 were combined to define ‘strong certainty’, 4–7 defined ‘some certainty’ and 1–3 defined ‘little certainty’. Given the focus on reaching ‘certainty’, some analyses were simplified to a dichotomy of strong certainty (8–10) or not (1–7).

Participants were additionally asked to reflect on two distinct periods, during or after medical school, and to answer the following question: Thinking of people who have influenced you in your medical career progression, how influential were the following in the direction your career took once you began your medical training? Possible answers were: patients, registrars, educator/mentor, consultant, other health professional. Participants reported on a 5-level scale, ranging from 1 (‘No influence’) through to 4 (‘Great influence’) and 5 (‘Determined my career path’), with the two highest categories (4–5) combined to define ‘strong influence’ or not (1–3).

### 2.4. Covariates

Other covariates for this study included gender, rural origin (defined as minimum six years childhood in a rural area), completing 1–2 clinical years of training predominantly in medium-sized regional hospital settings (henceforth termed ‘rural training’) and age at graduation from medical school.

### 2.5. Analyses

All analyses used Stata SE 15.1 for Windows (Stata Corp, College Station, Texas) and *p* < 0.05 for statistical significance. Descriptive statistics were used to present the basic rates of doctors reporting the key outcomes. Chi-squared tests were used to compare the proportion ‘certain’ of decisions about specialty and current practice location between comparison groups.

## 3. Results

There were 775 eligible participants, with key characteristics summarised in Table 1. The overall (crude) response rate was 17.1% (775/4540), which approximated to 36.1% after adjusting for ‘not opened’ invitations (as defined by Survey Monkey). Compared with all invited UQ graduates, the eligible participants were somewhat more likely to be recent graduates, female (51% versus 44%), rural trained participants (30% versus 26%) and working rurally (24% versus 20%).

### 3.1. Certainty of Specialising as a GP or Non-GP Specialist

When participants reflected about the timing of their specialty decision, the proportion reporting strong certainty varied enormously from the beginning of medical school through to the point of completing all specialty training (Figure 1). More GP specialists expressed certainty of their chosen specialty at the beginning of medical school (22% versus 12%), however, there were similar certainty levels at the end of medical school, with 27–29% having strong certainty of their specialty field and around half expressing some certainty. After internship, less than half continued to have strong certainty of their chosen specialty for both those choosing general practice (44%) or non-GP specialties (47%). At the start of their registrar training, more notable differences emerged, with around 65% of GPs compared with 80% of non-GP specialists having strong certainty of the specialty field they wanted to practice and this difference remained at 76% (GP specialists) versus 93% (non-GP specialists) at the end of their registrar training.

Additional analyses revealed that certainty about their commitment to a specialty field somewhat differed by career stage when assessed across key strata (binary groups) of childhood origin, rural training location, age at graduation and gender (Figure 2 and Appendix A). Notably, there were no significant differences in the level of certainty between GP specialists and non-GP specialists, at the end of medical school or after internship for any covariates. Across the registrar training period, many significant differences emerged but they mostly related to lower certainty amongst GP specialists rather than their childhood origin, rural training location, age and gender. One exception was for the group who completed rural training during medical school, with an equivalent proportion certain of their GP specialty decision as those choosing non-GP specialties (Figure 2).

### 3.2. Certainty of Practising in Rural or Metropolitan Location

When participants reflected about the timing of their practice location decision, the proportion reporting strong certainty also varied greatly from the beginning of medical school through to the point of completing all training (Figure 3). There were strong and consistent differences at each career time point between those currently working in rural or metropolitan locations, with significantly less of those working rurally having strong certainty of the location where they wanted to practice medicine. Only 29% of those working rurally currently had strong certainty of this decision at the end of medical school training, compared with 45% of the group working in metropolitan areas. The proportion of those who were certain only increased slightly between the start of medical school and end of internship for either practice location decision. At the start of registrar training, a significantly smaller proportion of those had strong certainty about rural work (53%), compared with strong certainty about metropolitan work (63%).

Additional analyses revealed that when assessed across key strata binary groups of childhood origin, rural training location, age at graduation and gender, significant differences were common and were mostly due to fewer of those working rurally who were certain about the location that they wanted to practice medicine at each career point (Figure 4 and Appendix A). One exception is the rural origin cohort choosing to work rurally, which has the highest certainty proportion at the start of registrar training (71%), compared with only 36% metropolitan origin who continue on to work rurally. Metropolitan origin doctors who were working rurally largely reached that decision by the end of their registrar training (65%), having risen from only 31% at the completion of internship. Participation in rural training was not associated with higher proportions being certain of their decision to work rurally, except for the end of registrar training. Clinical training in regional hospitals of one year versus two years was also tested, but there were no significant differences of certainty at all career points.

When location and specialty were examined in combination, there was a strong correlation between the timing of these decisions. For example, amongst those working rurally, at the end of medical school 54% who had strong certainty of their practice location also had strong certainty of their specialty; in contrast, only 19% of those without certainty of their practice location had strong certainty of their specialty. This pattern was consistent and significantly different at all career points, indicating that strong certainty of rural location requires also (first) having strong certainty of specialty for most participants.

### 3.3. Other Influences on Certainty of Decision(s) of Specialty and Practice Location

Figure 5 summarises the proportion of respondents influenced by different people over their early medical career. The influencers differed between GP and non-GP specialists. By far, the most common influence for all specialists was a consultant that they encountered after medical school for all doctors, although this was more common for non-GP specialists (61%) than GP specialists (35%). Non-GP specialists were also more strongly influenced by contact with registrars than GP specialists, in both medical school and afterwards. Those choosing general practice were slightly more influenced by an educator/mentor during medical school (23% vs. 19%), otherwise their career decisions were consistently less influenced by all the measured sources. In contrast, only minor differences in influences were observed between those working in rural or metropolitan areas. Most notably, more of those working rurally were influenced after medical school, by the consultants that they encountered (54% vs. 48%).

## 4. Discussion

This paper presents important empirical evidence about the emergence of certainty of choosing general practice and rural work over the early career stages that doctors progress through. This has important implications for workforce planning and addressing workforce distribution concerns. It strongly demonstrates that both choices may take many years to strengthen, thus contesting the long-term reliability of stated preferences or intent across medical school, which ignore the postgraduate stage of training. Generally, key factors of rural origin, participating in rural training during medical school, age at graduation and gender were marginally associated with more junior doctors reaching strong certainty of these career decisions at an earlier time point. However, the impact of time and the range of postgraduate training experiences, work, familial and environmental influences are also likely to play a part in affirming whether early career doctors grow in certainty about key career decisions [17,35]. Further, GP careers and rural careers seem to take longer to affirm decisions, compared with other specialties and metropolitan practice, and perhaps should not be rushed. This may relate to historically there being proportionally fewer clinical training experiences in smaller rural settings or with GPs, though Australia’s expansion of its national rural generalist pathway may assist [39]. Other research shows that generalists need time to experience a range of areas of clinical work that they like in order to decide their career [17]. Non-GP specialty decisions, unlike for GPs, were strengthened by positive experiences with registrars and consultants after medical school, and thus may require less time to navigate the decision to pursue a narrower field under key mentors which are often found in the hospital system.

Choosing a specialty is known to be a complex process, thus it is not surprising many junior doctors take a long time to reach strong certainty of their decision [17,40]. One notable finding is that junior doctors with rural clinical training experience were significantly more likely to be certain of their decision to choose general practice. It is unclear if this relates to rural training participants being more inclined to work rurally (and thus choose general practice to fit with that lifestyle) or perhaps from rural training experiences, including more time in general practice, connecting with more positive rural GP role models or enabling better integration of primary care influences. Global trends of declining preferences for primary care amongst new graduates suggest that many will need to be drawn back to primary care, possibly from ruling out other specialties first or placing more importance on the lifestyle benefits that are often associated with general practice [17,41]. This is supported by our evidence that strong certainty of GP decision consistently occurs at later career points to non-GP specialties. Additionally, there were fewer clear influences to confirming a GP decision, possibly relating to most training occurring in hospitals without many connection points to GPs or primary care itself. Moreover, undermining the status of GP as a specialty by non-GP specialists in hospital settings is known to diminish the attractiveness among recent graduates [17,40,42]. These likely relate to a slower decision point to choosing general practice for some.

Key global strategies to address the shortages of rural doctors include selecting more students with a rural origin and providing more clinical training in rural communities [26]. It is notable that the evidence demonstrating these to be associated with increased rural practice outcomes mostly comes from snapshots or single cross-sections of the workforce. As such, the process behind these key ‘clinical experiences’ leading individual doctors to rural practice is often simplified to a fixed effect size that applies to all such graduates, such as rural origin doctors being 2–3 times more likely than metropolitan-origin doctors to practise rurally [28]. Furthermore, evidence of rural practice uptake is usually divorced from specialty decisions. Our data demonstrate that most junior doctors had strong uncertainty of their practice location throughout their Australian training pathway, which is partly explained by uncertainty of their specialty decision. It is highly simplistic to assume that having these rural training experiences or early declarations of a rural intent/preference guarantees large numbers will stay rurally. The high level of uncertainty of their practice location can be seen as both a negative and a positive—there remains a large pool who are continually reassessing their career decisions and this includes those considering rural practice [43]; however, many may be open to the idea of rural practice but eventually decide otherwise, as competing interests, such as family needs, take preference [44]. This large pool of ‘fence sitters’ thus highlights the ongoing importance of maximising positive experiences amongst junior doctors of factors known to both pull towards and retain them in rural practice, whilst minimising negative experiences of factors known to push them away from rural practice [37,43]. The importance of continued opportunities for postgraduate rural training pathways to support growing the rural workforce has been demonstrated amongst one specialty group (international medicine) [45].

It is clear that major career decisions amongst junior doctors are far from being fixed outcomes, even after considering Australia’s somewhat later timeline of entering specialty training and ongoing uncertainty of their annual work location through to completion of their training. Strong certainty of working rurally only rises to above 50% by the start of their registrar training, with only 29% having strong certainty at the end of medical school. Evidence from North America suggests that post-matriculation factors, such as their spouse’s background, residency location and a rural postgraduate curriculum, add little predictive power of those working rurally and thus their presence is of low importance [46]. Our evidence instead suggests that confirmation of major career decisions occurs over a long period of time, with these later occurring factors likely to be contributing to the process of ruling in and then confirming their commitment to these key decisions [47].

Our study has several limitations. In addition to its low response rate, it relies on participants reflecting on their key career decisions, across multiple time points of up to 17 years duration. The results are likely to be impacted by recall bias. Recall of career decision certainty may have been better for more recent career stages. Current work location may not represent their long-term career decision, particularly amongst pre-registrars who are unlikely to have strong certainty of their decision yet. Whilst this study also collected their preferred location, we chose not to use this measure as its reliability of predicting actual future practice location, particularly after all training is completed, is questionable. These findings are also limited to one institution’s rural training program, which is mostly based in regional hospitals over one to two years, which may not be reflective of the results of immersion in a continuity-based program centred on primary care, such as a rural LIC. Finally, our options for key influences of career progression were limited to clinicians or health-related experiences, but this could be strengthened with the inclusion of known non-professional influences, such as family needs [48].

## 5. Conclusions

This study provides new evidence from one Australian medical school’s early-career graduates that their certainty of key career decisions tends to emerge and is affirmed late in their training. Certainty about a career in general practice and rural work takes longer to emerge than for counter options. The high proportion of uncertainty in early medical careers of both decisions highlights the importance of regular positive experiences throughout both under and postgraduate stages of training to reinforce orientation towards general practice and rural work. This could build on other key strategies, such as selecting doctors with a rural background and rural clinical training.

## Figures and Tables

**Figure 1 ijerph-18-11835-f001:**
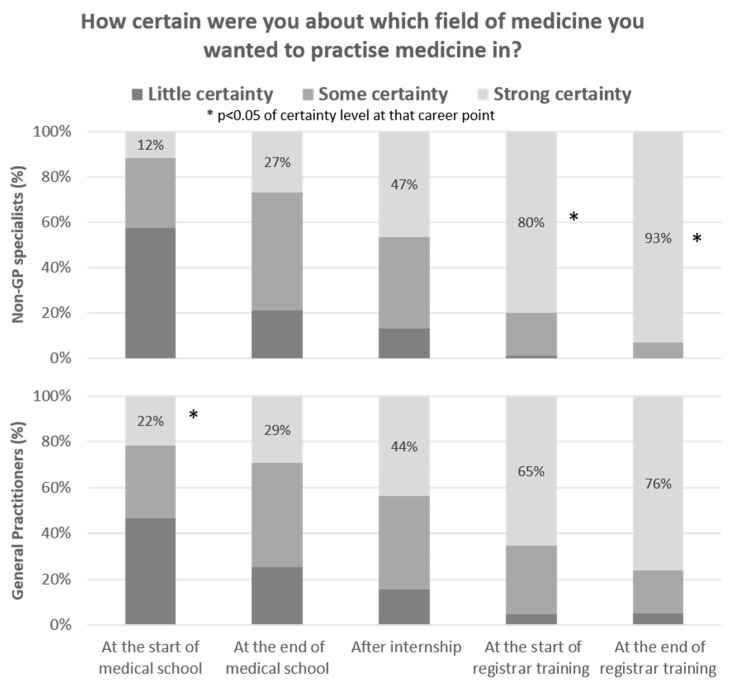
Level of certainty of their specialty decision, by junior doctors’ reflection on five career points.

**Figure 2 ijerph-18-11835-f002:**
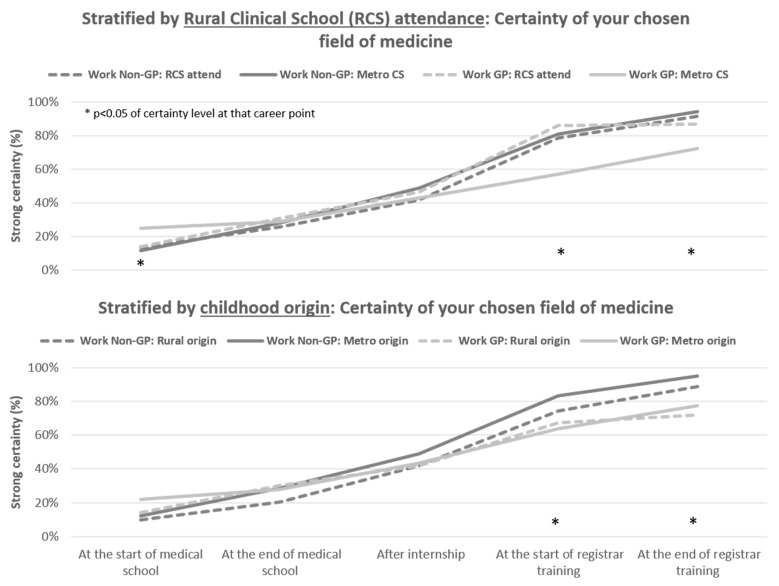
Level of certainty of their specialty decision amongst junior doctors, by key rural strata (childhood origin, clinical school training location).

**Figure 3 ijerph-18-11835-f003:**
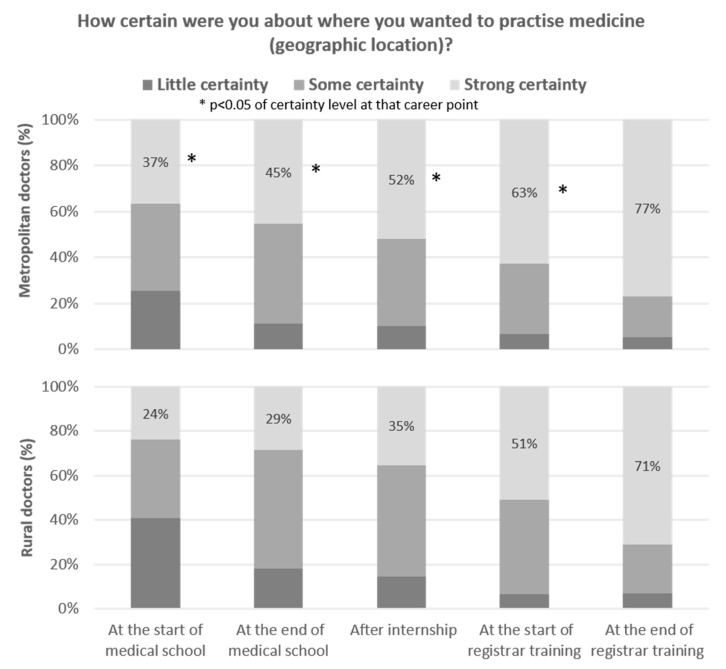
Level of certainty of their practice location decision, by junior doctors’ reflection on five career points.

**Figure 4 ijerph-18-11835-f004:**
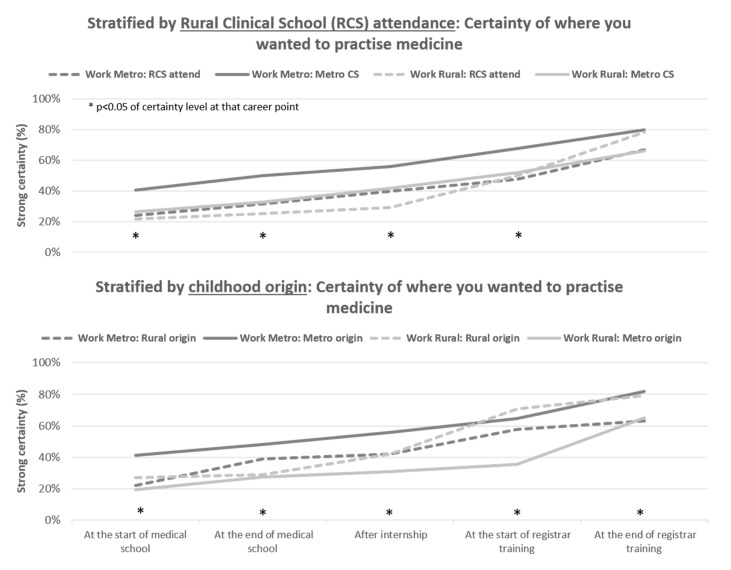
Level of certainty of their practice location decision amongst junior doctors, by key rural strata (childhood origin, clinical school training location).

**Figure 5 ijerph-18-11835-f005:**
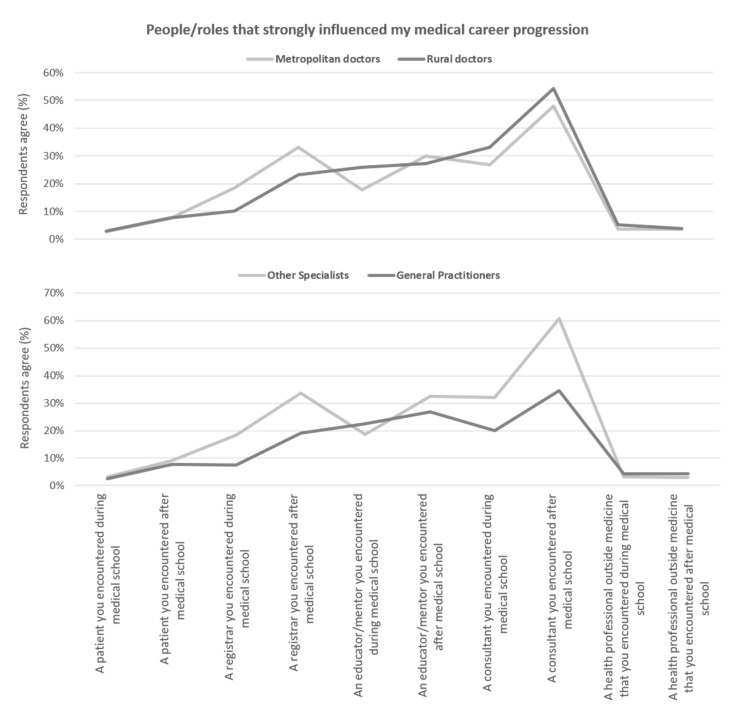
Self-identified people or experiences that, on reflection by junior doctors, strong influenced their career progression decision(s) of specialty and practice location.

**Table 1 ijerph-18-11835-t001:** Summary of participants in the medical graduate outcomes study (2019), compared against all invited graduates.

Variable	Group	Participants	All UQ Graduates
PGY	1–3	172 (22%)	939 (21%)
	4–6	193 (25%)	896 (20%)
	7–9	166 (21%)	884 (19%)
	10–13	145 (19%)	976 (21%)
	14–17	99 (13%)	845 (19%)
Gender	Male	378 (49%)	2542 (56%)
	Female	397 (51%)	1998 (44%)
RCS participants	Yes	236 (30%)	1191(26%)
	No	539 (70%)	3349 (74%)
Age at medical school graduation	28+	223 (29%)	1242 (27%)
	Under 28	552 (71%)	3298 (73%)
Working rurally (MMM 2–7)	Yes	176 (24%)	899 (20%)
	No	573 (76%)	3558 (80%)

PGY: Post-graduate year (number of years since graduating from medical school); RCS = Rural clinical School; MMM = Modified Monash Model (7-level rurality classification).

## Data Availability

The data presented in this study are available on request from the corresponding author. The data are not publicly available due to ethical restrictions.

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
