# Peer review of "Exploring Doctors’ Emerging Commitment to Rural and General Practice Roles over Their Early Career"

_ijerph, 2021, doi:10.3390/ijerph182211835_

Round 1

Reviewer 1 Report

Overall interesting and sufficiently innovative research adding upon some partly outdated scientific findings. The methodological approach appears to be adequate although the relatively low return reduces the strength of the finding, particularly because the correctly mentioned recall biased is likely to add upon a selection bias. Would have been interesting to stratify more clearly according to the period of time passed since the graduation.

The findings are presented correctly but would require a bit more of an explanation particularly for those readers who are not intensively informed about the topics and the Australian medical training system. The figures don't work in a black-and-white printout and should be better adjusted to this mode of printing.

The authors refer partly to slightly outdated literature, and they particularly question findings published more than 10 years ago. The state of research, however, has meanwhile developed and some more complex analyses are meanwhile available which the authors might refer to and compare there findings with.

Author Response

Point 1: Overall interesting and sufficiently innovative research adding upon some partly outdated scientific findings. The methodological approach appears to be adequate although the relatively low return reduces the strength of the finding, particularly because the correctly mentioned recall biased is likely to add upon a selection bias. Would have been interesting to stratify more clearly according to the period of time passed since the graduation.

Response 1: One of the difficulties of further stratification of ‘time since graduation’ is that we don’t readily know the specific time point that they either started their registrar training or ended their registrar training. Firstly, these time points can vary enormously in the Australian training pathway and secondly, we did not specifically request these measures. Whilst we can examine associations between different PGY groups (as per Table 1), these do not readily align with time since fellowship, etc. In addition, we don’t have a strong hypothesis to explore this further. For interest only, we tested stratifying certainty of (1) field of medicine; (2) geographic location at the end of registrar training by their ‘years since graduation’ (PGY), but observed patterns were inconsistent (non-significant).

Point 2: The findings are presented correctly but would require a bit more of an explanation particularly for those readers who are not intensively informed about the topics and the Australian medical training system. The figures don't work in a black-and-white printout and should be better adjusted to this mode of printing.

Response 2: We apologise for overlooking the ability to print in black and white. All figures have been edited to address this. In line with another reviewer’s request, additional information of the Australian PG medical training system (and UQ’s training) has been added to the Methods, to aid its readability for the global audience. We have expanded our explanation of the results in the Discussion sections, where relevant noting these against the Australian context.

Point 3: The authors refer partly to slightly outdated literature, and they particularly question findings published more than 10 years ago. The state of research, however, has meanwhile developed and some more complex analyses are meanwhile available which the authors might refer to and compare their findings with.

Response 3: We agree that the evidence base around rural medical workforce has strengthened greatly, including some emerging studies using more complex analyses. We have drawn from much of this literature, with 27/42 references being from 2017-21. However, a key distinction is that most evidence still relates to factors associated with rural (or generalist) practice, without consideration of the process or timeframe behind that complex decision-making chain. We had focused on Rabinowitz (2012) because it openly questions the ‘contribution’ of postgraduate factors after medical school graduation in their career choice. We have edited this section so that its USA context is noted and our argument is less focused on contesting their viewpoint as if it is was ‘gold standard’ evidence.

Reviewer 2 Report

Dear authors,

Thanks for sending this article for review.

This is a cross sectional survey of 775 medical graduates; low response rate is to be anticipated. Participants were asked to response re rurality of origin, training and current employment, specialty and certainty about decision by point in time. 

The paper tracks workplace influences on decision regarding career choice - primary care vs specialty and re rurality.

I presume authors have also asked re age, gender as these would be important to analyse; perhaps another paper is planned. Also whether each person has a partner/ family may be a predictor - again, would be interesting to see if authors considering work from the dataset on this.

Analysis is appropriate for a cross sectional study, with no higher analysis, however the paper produces some interesting findings re timeliness of influences on career choice. - recommend publish.

Figures well presented and clear to the reviewer. Figure 5 was useful to me as an educator.

References  typo ref 7

Author Response

Point 1: I presume authors have also asked re age, gender as these would be important to analyse; perhaps another paper is planned. Also whether each person has a partner/ family may be a predictor - again, would be interesting to see if authors considering work from the dataset on this.

Response 1: The reviewer may not have seen these, but in the Supplementary Materials we submitted additional results stratified by Age at graduation and Gender. They are briefly referenced in the text in the Results section below Figure 1 and Figure 3. Findings relating to these 2 factors are also briefly noted already in the Discussion (paragraph 1). Minimal clear patterns emerge, so we decided they better belonged as supplementary figures.

Point 2: Analysis is appropriate for a cross sectional study, with no higher analysis, however the paper produces some interesting findings re timeliness of influences on career choice. - recommend publish. Figures well presented and clear to the reviewer. Figure 5 was useful to me as an educator. References typo ref 7

Response 2: Thankyou for your review. Reference 7 has been edited.

Reviewer 3 Report

The authors are to be congratulated on their initiative in undertaking and reporting this important research. The study design is sound and the presentation of results are clear and understandable. There is sufficient information for readers to be able to replicate this research in their own setting.
Having said that, I am concerned at the authors' somewhat superfial interpretation of the literature regarding the complexity of undergraduate and postgraduate medical education and medical career pathways, including pathways to rural practice. For example, they use the terms "exposure" and "immersion" without clear definitions or citing of relevant medical education literature, such as the importance of "continuity" in patient contact, curriculum and clinical teachers. There is no mention of the literature on longitudinal integrated clerkships (LIC).
These issues become relevant when there is recognition of the specific University of Queensland (UQ) curriculum that is specialist dominated and has no LIC. In fact, there is no acknowledgement by the authors that the findings have potential limitations in interpretation, not only from a methodological perspective because of the very low response rate, but also because of the specifics of the UQ curriculum. Rural Clinical School (RCS) appears to be considered by the authors to be "rural immersion" although the students have most of their clinical education in large regional hospitals with specialists as their principal clinical teachers and role models, rather than general practitioners. The article would be improved by a brief description of UQ curriculum in the Introduction, and then acknowledgement of those limitations in the Discussion.
Another methodological issue is the lack of specific explanation and justification for the items that were included in the survey instrument. This is in addition to the lack of acknowledgement that prevocational training positions are predominantly metropolitan and regional hospital-based with limited if any clinical experience/training in smaller hospitals or general practice.
The Discussion would be improved by a description and acknowledgement of the fragmented and complicated nature of junior doctors' situation in Australia. This is described well in the Health Workforce Australia 2014 report (Health Workforce Australia. Australia’s future health workforce - doctors). It is particularly challenging for junior doctors having to apply for and compete for their jobs every year at both prevocational and vocational training levels. For many of these doctors, this also coincides with major life-cycle developments including family issues. These contributors to "uncertainty" ought to be acknowledged by the authors. In North America, the year by year application requirement does not exist.
Speaking of North America, the statement by the authors that "our evidence strongly contests..." with citation of only one reference from 2012 is at best an exaggeration, and at worst a sign that the authors lack the depth of understanding of the different contexts in different countries, including training systems.
In summary, I see this as important research that justifies publication in this Journal. Having said that, there is a need for the authors to demonstrate a depth of understanding of the medical education/training journey, including the limitations of a study involving just one medical school in one country, both in the Introduction and in the Discussion sections to be able to justify a clear Conclusion.

Author Response

The authors are to be congratulated on their initiative in undertaking and reporting this important research. The study design is sound and the presentation of results are clear and understandable. There is sufficient information for readers to be able to replicate this research in their own setting.

Thank you for noting the positive aspects of our research.

Point 1: I am concerned at the authors' somewhat superficial interpretation of the literature regarding the complexity of undergraduate and postgraduate medical education and medical career pathways, including pathways to rural practice. For example, they use the terms "exposure" and "immersion" without clear definitions or citing of relevant medical education literature, such as the importance of "continuity" in patient contact, curriculum and clinical teachers. There is no mention of the literature on longitudinal integrated clerkships (LIC).

Response 1: Thank you for highlighting this. We are acutely aware of the complexity of medical training and career pathways and the need to define terminology to ensure consistent global interpretation. We have added a few sentences to paragraph 5 relating to the different types of rural medical education programs, including LICs which our team has been involved in studying as part of the global LIC consortium. We have clarified the text about rural immersion, how long we mean when we say this, and also about experience or exposure that doctors get within LICs and what this encompasses.

Point 2: These issues become relevant when there is recognition of the specific University of Queensland (UQ) curriculum that is specialist dominated and has no LIC. In fact, there is no acknowledgement by the authors that the findings have potential limitations in interpretation, not only from a methodological perspective because of the very low response rate, but also because of the specifics of the UQ curriculum. Rural Clinical School (RCS) appears to be considered by the authors to be "rural immersion" although the students have most of their clinical education in large regional hospitals with specialists as their principal clinical teachers and role models, rather than general practitioners. The article would be improved by a brief description of UQ curriculum in the Introduction, and then acknowledgement of those limitations in the Discussion.

Response 2: Similar to >90% of the literature in rural training program outcomes, we agree that our evidence is limited by the findings relating to graduates of a single institution. We have added a statement to the limitations section. However, this is also a strength, because the training program characteristics are relatively consistent. To assist this, we have added a section to the Methods (context) outlining the rural training characteristics at UQ.

Point 3: Another methodological issue is the lack of specific explanation and justification for the items that were included in the survey instrument.

Response 3: We have added two statements under the Methods. Firstly, we have clarified that the survey instrument was purposefully designed to meet our broader project’s objective to better understand the career decision making process and outcomes of UQ medical graduates. Secondly, we note that the chosen 5 career points align with key distinctive points in the Australian training context.

Point 4: This is in addition to the lack of acknowledgement that prevocational training positions are predominantly metropolitan and regional hospital-based with limited if any clinical experience/training in smaller hospitals or general practice.

Response 4: Firstly, in paragraph 1 of the Discussion we have added a statement clarifying that delays to affirming career decisions in rural and/or general practice may relate to there being proportionally fewer training exposures in smaller rural settings or with GPs. Secondly, we have added detail under Context of this study that there are fewer pre-registrar opportunities in smaller rural hospitals and/or outside of the hospital setting such as general practice.

Point 5: The Discussion would be improved by a description and acknowledgement of the fragmented and complicated nature of junior doctors' situation in Australia. This is described well in the Health Workforce Australia 2014 report (Health Workforce Australia. Australia’s future health workforce - doctors). It is particularly challenging for junior doctors having to apply for and compete for their jobs every year at both prevocational and vocational training levels. For many of these doctors, this also coincides with major life-cycle developments including family issues. These contributors to "uncertainty" ought to be acknowledged by the authors. In North America, the year by year application requirement does not exist.

Response 5: This was an oversight to provide insufficient detail of the Australian training context. We agree with most of the points raised of the lack of strong pathways to rural and/or generalist careers, something we have recently published findings about. We have added a section to the Methods (context) relating to describing the Australian training context. This contextualisation is briefly expanded upon in the Discussion of the findings.

Point 6: Speaking of North America, the statement by the authors that "our evidence strongly contests..." with citation of only one reference from 2012 is at best an exaggeration, and at worst a sign that the authors lack the depth of understanding of the different contexts in different countries, including training systems.

Response 6: We had focused on Rabinowitz (2012) because it openly questions the ‘contribution’ of postgraduate factors after medical school graduation in their career choice. We have edited this section so that its USA context is noted, we have removed statements like “strongly contests”, and our argument is less focused on contesting their viewpoint as if it is was ‘gold standard’ evidence.

Point 7: In summary, I see this as important research that justifies publication in this Journal. Having said that, there is a need for the authors to demonstrate a depth of understanding of the medical education/training journey, including the limitations of a study involving just one medical school in one country, both in the Introduction and in the Discussion sections to be able to justify a clear Conclusion.

Response 7: In line with their earlier points raised, we have added detail of the Australian and UQ context to enable a clearer statement of the generalisability of this study’s findings. Through the Discussion and Conclusion, we have added a number of statements that such evidence presented is drawn from these specific contexts.

Round 2

Reviewer 3 Report

I thank the authors for their clear and  specific responses to the concerns that I raised. Overall, the changes that the authors made address these concerns satisfactorily.

Having said that, I still have some misgivings regarding the authors' use of the term "rural immersion" for a year (or two) of hospital specialty block rotations in the regional centres that are the base sites for the UQ Rural Clinical School. In part, this concern relates to the perception that the regional centres are not really "rural". Also, the authors did not pick up on my comments regarding the importance of "continuity" in patient contact, curriculum and clinical teachers (see Hirsh DA, Ogur B, Thibault GE, Cox M. 2007. “Continuity” as an Organizing Principle for Clinical Education Reform. N Engl J Med 356;8: 858-866, and many subsequent articles on LICs). This is a critical component of LICs and contributes to true rural immersion in which students are living and learning in rural communities and their intense interactions with patients guide the students' learning and professional identity formation. In this article, I suggest replacing "rural immersion" with "prolonged Rural Clinical School placement (or attachment)".

In this context, I see "immersion" as prolonged living and learning in rural communities and different from short term "exposure" of 4-6 week rural placements. Consequently, I am not comfortable with "immersive as a rural exposure" in line 89-90. In fact, I suggest that the authors review the use of the term "exposure(s)" throughout the manuscript. In many cases, I suggest "clinical experience" or "clinical training".  

Author Response

I thank the authors for their clear and specific responses to the concerns that I raised. Overall, the changes that the authors made address these concerns satisfactorily.

Point 1:

Having said that, I still have some misgivings regarding the authors' use of the term "rural immersion" for a year (or two) of hospital specialty block rotations in the regional centres that are the base sites for the UQ Rural Clinical School. In part, this concern relates to the perception that the regional centres are not really "rural".

Response 1:

On the one hand, we wholly agree with the reviewer in that the 1-year or 2-year placements described in this study are more correctly termed ‘regional’, rather than ‘rural’. On the other hand, it is very common in both policy and the published literature to use the term ‘rural’ as the catch-all term that includes all locations not considered to be metropolitan (or ‘major city’) areas. Australia’s national training program in these geographical areas come under ‘Rural Clinical Schools’. Moreover, throughout the paper we use the phrases ‘rural practice’ or ‘working rural[ly]’ as inclusive of regional centres, smaller rural and remote locations.

Based on points 2 and 3 below, we understand that the reviewer’s greater concern is our use of the phrase ‘rural immersion’. We agree and have removed its use from the paper. Our updated definition(s) are exemplified in our statement under section 2.4, where we have deleted ‘rural immersion’ and clarified in this paper that ‘rural training’ refers to “completing 1-2 clinical years of training in predominantly medium-sized regional hospital settings”.

Point 2:

Also, the authors did not pick up on my comments regarding the importance of "continuity" in patient contact, curriculum and clinical teachers (see Hirsh DA, Ogur B, Thibault GE, Cox M. 2007. “Continuity” as an Organizing Principle for Clinical Education Reform. N Engl J Med 356;8: 858-866, and many subsequent articles on LICs). This is a critical component of LICs and contributes to true rural immersion in which students are living and learning in rural communities and their intense interactions with patients guide the students' learning and professional identity formation. In this article, I suggest replacing "rural immersion" with "prolonged Rural Clinical School placement (or attachment)".

Response 2:

We strongly agree that LICs in smaller rural locations can be a critical model for student learning. We apologise if our use of the phrase “rural immersion” was in any way inferring that this study’s clinical / training experiences were similar to those of a rural LIC. As per point 1, we have wholly removed the phrase ‘rural immersion’ from the paper. Additionally, we have noted the ‘continuity’ aspect of the rural LIC model in both the Introduction and the limitations.

Point 3:

In this context, I see "immersion" as prolonged living and learning in rural communities and different from short term "exposure" of 4-6 week rural placements. Consequently, I am not comfortable with "immersive as a rural exposure" in line 89-90. In fact, I suggest that the authors review the use of the term "exposure(s)" throughout the manuscript. In many cases, I suggest "clinical experience" or "clinical training".

Response 3:

The noted phrase “immersive as a rural exposure” has been deleted, in line with our responses to points 1 and 2 above.

Additionally, where we had previously used the word ‘exposure’ to relate specifically to clinical training, we have rephrased these as ‘clinical [training] experiences’ or ‘rural training experiences’.